# Construction and investigation of multi-enzyme immobilized matrix for the production of HFCS

**Sabbir Janee[1], Shatabdy Saha[1], Sabrina Sharmin[2], A. Q. Fuad Hasan[1], Umme Salma Zohora[1], Ripa Moni[1], Md. Zahidul Islam◯[1]\*, Mohammad Shahedur Rahman◯[1,3]**

**1** Department of Biotechnology and Genetic Engineering, Jahangirnagar University, Savar, Dhaka, Bangladesh, **2** School of Pharmacy, BRAC University, Dhaka, Bangladesh, **3** Wazed Miah Science Research Center, Jahangirnagar University, Savar, Dhaka, Bangladesh

\* zahidul@juniv.edu

**Data Availability Statement:** All relevant data are within the paper and its Supporting Information files.

## Abstract

Enzymes are biological molecules that act as catalysts and speed up the biochemical reactions. The world's biotechnological ventures are development of enzyme productiveness, and advancement of novel techniques for thriving their shelf existence. Nowadays, the most burning questions in enzyme technology are how to improve the enzyme productivity and reuse them. The immobilization of enzymes provides an excellent scope to reuse the enzymes several times to increase productivity. The main aim of the present study is the establishment of an immobilized multi-enzyme bio-system engineering process for the production of High-fructose corn syrup (HFCS) with an industrial focus. In this study, multi-enzyme such as α-amylase, glucoamylase and glucose isomerase were immobilized in various support matrices like sodium alginate, sawdust, sugarcane bagasse, rice bran and combination of alginate with cellulosic materials. The activities of the immobilized multi-enzyme system for the production of HFCS from the starch solution were determined. The multi-enzyme immobilized in sodium alginate shows better fructose conversion than free enzyme. Among the support matrices, multi-enzyme immobilized in sawdust produced total 80.74 mg/mL of fructose from starch solution and it was able to be used in several production cycles. On the other hand, multi-enzyme immobilized in combination of sodium alginate and sawdust produced the maximum amount of fructose (total 84.82 mg/mL). The free enzyme produced 17.25 mg/mL of fructose from the starch solution in only a single cycle. In this study a new fixed bed immobilized multi-enzyme bioreactor system was developed for the production of HFCS directly from starch. This finding will create a new opportunity for the application of immobilized multi-enzyme systems in many sectors of industrial biotechnology.

## 1. Introduction

The enzyme is a protein that acts as a natural catalyst; it speeds up reactions by decreasing the activation energy, without changing its own molecular structure. Enzymes are exceedingly

**Funding:** The author(s) received no specific funding for this work.

**Competing interests:** The authors have declared that no competing interests exist.

precise, catalyzing a single biochemical reaction or a very few closely related reactions [1]. The actual structure of an enzyme and its active site delimit the specificity of the enzyme. Enzymes can often perform reactions which are not even feasible with conventional chemistry and thereby provide novel benefits. The enzyme catalyst reaction is capable of accelerating a chemical reaction without being part of the final product or being consumed in the process [2].

Enzyme α-amylase is a hydrolase enzyme that catalyzes the initial hydrolysis of corn starch into shorter oligosaccharides through and cleaves α-D-(1–4) glycosidic bond [3]. The end products of α-amylase action are oligosaccharides, which is a mixture of maltose, maltotriose, dextrin and branched oligosaccharides of 6–8 glucose units that contain both $\alpha$-1,4 and $\alpha$-1,6 linkages [4]. Enzyme α-amylase can be widely found in microorganisms, plants and higher organisms [5].

Glucoamylase is a hydrolase enzyme that successively hydrolyzes α-1, 4 and α-1, 6 glycosidic bonds consecutively at the non-reducing end of starch molecules and releases free glucose [6]. Glucoamylse is often described separately from amylase because it digests starch and dextrin by removing a glucose molecule from the end of polysaccharides rather than cleaving longer strings of glucose molecule in the middle, forming small chains [7]. Glucoamylase can derive from a wide variety of plants, animals and microorganisms, but most glucoamylases occur in fungi. In the prospect of commercialization glucoamylase originates from strains of either *Aspergillus niger* or *Rhizopus sp.* [8].

Glucose isomerase is an isomerase enzyme that catalyzes the conversion of glucose to fructose. Glucose isomerase is generally produced in prokaryotes. Various bacteria and actinomycetes produce glucose isomerase [9]. The majority of them is intracellular enzymes. It has been accounted for that just a couple of organisms produce extracellular glucose isomerase enzymes [10, 11].

High fructose corn syrup (HFCS) is produced from corn starch to yield fructose and glucose mixture, there are three essential enzymes involved in the HFCS making process: (i) α-amylase helps to break the large starch molecule to maltodextrin by liquefaction, (ii) glucoamylase helps to convert maltodextrin to glucose by scarification, (iii) glucose isomerase converts glucose to fructose by isomerization [12]. The combination of all three enzymes (α-amylase, glucoamylase and glucose isomerase) converts corn starch to HFCS.

All three enzymes are unstable in normal condition [13]. In industrial application it is often impeded by a lack of long-term operational stability, and it is challenging to the recovery process and reuses all the enzymes for the further chemical reaction [14]. In order to make enzymes more effective in industrial perspective, different methods of cost reduction have been put into practice and multi-enzyme immobilization is one of them. Multi-enzyme immobilization is a process in which different types of enzymes are immobilized in a single matrix [15]. Multi-enzyme immobilization provides several benefits, for example (i) rapid arrest of the reaction between enzymes and substrate (ii) increase enzymes permanency against temperature, solvents, pH, contaminants and impurities () recovery and reuse of expensive enzymes (v) permits their application in continuous fixed-bed bioreactor (v) reduce downstream product processing. Enzymes are attached with a supporting matrix through adsorption, microencapsulation, cross-linking, ionic bonding and covalent bond [16]. However, the use of common supporting matrix for multi-enzyme immobilization namely, sodium alginate, cellulosic materials, silica-based carriers, acrylic resins, synthetic polymers, active membranes and exchange resins faces [17]. Multi-enzyme immobilization process easily carried out through sodium alginate but dependent on industry focus, this process is not effective and reliable. Cellulosic materials are very cheap and can be easily found in the local environment. The cellulosic materials carry many hydroxyl groups that allow binding the enzymes on its surface. Oxidation process helps to insert a number of aldehyde groups into cellulosic materials that

covalently bind to the amino group of enzymes [4]. The change of surface morphology of cellulosic materials before oxidation process and after oxidation process can be observed through SEM (scanning electron microscope) [18].

Generally, most of the enzymes are comparatively unusable after reaction and they have high separation costs after use. Nowadays immobilized enzymes have received greater attention in different sectors especially in the food industry. Enzyme immobilization is a way of confining the enzyme molecules to a solid support over which a substrate is passed and converted to product. On the other hand, Multi-enzyme immobilization is a process in which different types of enzymes are immobilized in a single matrix. Multi-enzyme immobilization has attracted a wide range of interest from fundamental academic research to many different industrial applications such as pharmaceutical industry, food industry, beverage industry, and other chemical industry.

We use sucrose as table sugar in our country. HFCS is a liquid alternative sweetener of sugar that is made from corn starch or other starch producing grains, using various enzymes such as α-amylase, glucoamylase and glucose isomerase. If fructose syrup can be produced from starch using an immobilized method, the cost will be reduced. Corn is an important cereal in Bangladesh. Therefore, it is very relevant to produce HFCS from corn starch in Bangladesh. HFCS is 1.3 times sweeter than sucrose and 2.3 times sweeter than glucose. HFCS is also safe for diabetes patients because it is metabolized in the liver but it releases energy slower than glucose [19].

In Bangladesh, application of multi-enzyme immobilization in industrial scale is not commonly seen. However, it will be very much economical, if we can establish a bio-system engineering process based on immobilize enzymes for HFCS production. In this study, multi-enzyme (α-amylase, glucoamylase and glucose isomerase) were immobilized in alginate beads and cellulosic material like sawdust, sugarcane bagasse and rice bran for the production of the HFCS from the corn starch. A fixed bed bioreactor has been developed to produce the HFCS. The present study will help to understand the immobilization process of multiple enzymes, optimization parameters and comparison between immobilized and free enzymes during the production of HFCS.

## 2. Materials and methods

### 2.1 Reagents

In this experiment the chemicals and reagents have been chosen in such a way that is cost effective and serve the purpose well. Also, the reagents have been chosen according to the purpose, quality, price and availability. Enzymes α-amylase, glucoamylase and glucose isomerase were purchased from Sigma (Switzerland), Tokyo chemical industry (Japan) and Noor enzyme (India) respectively. Corn starch, D-fructose, sodium metaperiodate, $H_2SO_4$, NaOH, NaCl, HCl and $CaCl_2$ were purchased from Merck (Germany). Other reagents like sodium metaperiodate and resorcinol were purchased from Merck (India) and Loba Chemical (India) respectively. The cellulosic materials such as sawdust, sugarcane bagasse and rice bran were collected from local markets (Savar, Dhaka, Bangladesh).

### 2.2 Sodium alginate bead preparation

The powder form of all enzymes dissolved in distilled water. The concentration of α-amylase, glucoamylase and glucose isomerase were 0.002 mg/mL, 0.002 mg/mL and 21 mg/mL, respectively. In case of multi-enzyme immobilization, 0.2 mg of α-amylase, 0.2 mg of glucoamylase, and 2100 mg of glucose isomerase enzymes were dissolved in distilled water and make up volume 100 mL. Sodium alginate (0.5g) was dissolved in 10 mL of multi-enzyme solution and

stirred for 30 min using a magnetic stirrer. Alginate beads were prepared by dripping the multi-enzyme alginate solution onto 0.2 M CaCl$_2$ using a 10 mL syringe. Spherical beads were formed immediately after dripping the alginate solution into CaCl$_2$ solution. The beads were kept in CaCl$_2$ solution for 30 min without any disturbances. The beads were then separated by using filter paper and washed properly two times with distilled water. The beads were soaked in distilled water and stored at 4˚C for preservation.

In case of single-enzyme immobilization, 0.2 mg of α-amylase, 0.2 mg of glucoamylase, and 2100 mg of glucose isomerase enzyme were taken in different volumetric flasks and dissolved in distilled water and make up volume 100 mL of each flask. Sodium alginate (0.5g) was dissolved in 10 mL of α-amylase enzyme solution and stirred for 30 min using a magnetic stirrer. Alginate beads were prepared by dripping the α-amylase enzyme alginate solution onto 0.2 M CaCl$_2$ using a 10 mL syringe. Spherical beads were formed immediately after dripping the alginate solution into CaCl$_2$ solution. The beads were kept in CaCl$_2$ solution for 30 min without any disturbances. The beads were then separated by using filter paper and washed properly two times with distilled water. The same procedure was repeated for glucoamylase and glucose isomerase enzymes. Then different types of enzymes containing beads were soaked in distilled water and stored separately at 4˚C for preservation.

## 2.3 Determination of activity of different enzyme immobilization systems

For single-enzyme immobilization, each matrix material (Immobilized single-enzymes in sodium alginate) were taken in three different falcon tubes. 10 mL of corn starch solution (1%) was added in the tube containing the α-amylase immobilized matrix and incubated for 30 min at 37˚C. Then the solution was collected and added in the second tube containing the glucoamylase immobilized matrix and incubated for 30 min at 37˚C. Then the solution was collected from the second tube and added in the third tube containing the glucose isomerase immobilized matrix and incubated for 30 min at 37˚C. In case of multi-enzymes immobilization, the immobilized multi-enzyme in sodium alginate beads were taken in a falcon tube and 10 mL of corn starch solution (1%) was added in the tube. For the control experiment, 10 mL of corn starch solution (1%) and 10 mL free enzyme were taken in a falcon tube. All the tubes were incubated for 30 min at 37˚C. Then 1 mL solution from each tube was taken into new test tubes, and 2 mL of resorcinol solution was added and incubated all test tubes at 80˚C for 10 min in a water bath. The solution temperature was cooled at room temperature and optical density (OD) of control and sample solutions were measured at 520 nm wavelength using UV-Vis spectrophotometer. The productivity of fructose (mg/mL) was determined from a standard calibration curve. To determine the productivity of fructose by an immobilized enzyme in support matrices and free enzyme, three independent experiments were carried out for each matrix. Average values and standard deviations of productivity of immobilized enzymes in support matrices and free enzymes were calculated.

## 2.4 Collection and purification of cellulosic materials

Different cellulosic materials such as sawdust, sugarcane bagasse and rice bran were collected from the local market. All the materials were washed properly several times with tap water. Then they were washed with distilled water for 2 to 3 times. In the case of sugarcane bagasse, it was cut into very small pieces. All the materials were dried in a hot air oven for 48 to 72 h or until drying properly. After drying they were grinded finely using a grinder and stored in the refrigerator at 4˚C [4].

## 2.5 Modification of cellulosic materials

The purified samples (5 g) of each were taken into different flasks. For oxidation, 375 mL of 0.03 M periodic acid (sodium metaperiodate and sulfuric acid) was added in 5 g of each dried cellulosic material. Then the pH of the solution was adjusted to 3.0 using sulfuric acid. These solutions were kept in a shaking incubator at 60°C and 150 rpm for 24 h. After 24 h these materials were filtered and washed with distilled water several times [20]. After completion of washing, the three materials were transferred in separate petri plates, and then dried in a hot air oven for 48 to 72 h.

## 2.6 SEM visualization

The dried samples were taken in the Scanning Electron Microscope (SEM) chamber to observe the surface topography.

## 2.7 Immobilization of multi-enzyme in cellulosic materials

0.5 g of each dried cellulosic material (rice bran or sawdust or sugarcane bagasse) before and after oxidation treatment were immersed in 10 mL of multi-enzyme (α-amylase, glucoamylase and glucose isomerase enzymes) solution separately and incubated at 37°C in a shaking incubator for 1 h. The samples were filtered with filter paper to remove excess enzymes, and it was followed by washing two times with distilled water. Then the samples were taken into clean falcon tubes and stored at 4°C for further use.

## 2.8 Determination of HFCS

Each matrix material (Immobilized multi-enzymes in different materials such as sawdust, sugarcane bagasse, rice bran before and after oxidation and sodium alginate beads) were taken in different falcon tubes and 10 mL of corn starch solution (1%) was added in each tube. For the control experiment, 10 mL of corn starch solution (1%) and 10 mL free enzyme were taken in a falcon tube. All the tubes were incubated for 30 min at 37°C. Then 1 mL solution from each tube was taken into new test tubes, and 2 mL of resorcinol solution was added and incubated all test tubes at 80° C for 10 min in a water bath. The solution temperature was cooled at room temperature and optical density (OD) of control and sample solutions was measured at 520 nm wavelength using UV-Vis spectrophotometer. To determine the productivity of HFCS by an immobilized multi-enzyme in support matrices, three independent experiments were carried out for each matrix. Average values and standard deviations of productivity of immobilized multi-enzyme in support matrices were calculated.

## 2.9 Reusability test of immobilized multi-enzyme in different support materials

After completing the first cycle, immobilized multi-enzymes in the cellulosic materials, and sodium alginate beads were recovered and washed properly several times with distilled water. Then enzyme matrices were taken into different new falcon tubes and 10 mL of 1% starch solution was added into each falcon tube. All the tubes were incubated at 37°C for 30 min. Then 1 mL solution from each tube was taken into new test tubes, and 2 mL of resorcinol solution was added and incubated all the test tubes at 80° C for 10 min in a water bath. The solution temperature was cooled at room temperature and optical density (OD) of control and sample solutions was measured at 520 nm wavelength. After the completion of the second cycle, immobilized multi-enzymes matrices were recovered by filtration and washed properly several times. Above procedure was repeated for the third cycle, fourth cycle and so forth.

## 2.10 Preparation of different types of fixed bed multi-enzyme bioreactor system

A fixed bed bioreactor for the conversion of corn starch to HFCS was prepared using immobilized multi-enzymes. The bioreactor contains corn starch resolver, flow controller, support matrix, the filter foam sheet and product resolver [4]. To prepare fixed bed enzymes bioreactor systems, seven syringes (50 mL) were taken. Seven different types of multi-enzymes immobilized matrices were prepared. First, 0.5 g of sodium alginate was taken in 10 mL of multi-enzyme solution. Second, 1 g of purified and oxidized sawdust was taken in 10 mL of multi-enzyme solution. Third, 0.5 g of sodium alginate and 0.5 g of sawdust was taken in 10 mL of multi-enzyme solution and dissolved. Then 10 mL of 0.2 M $CaCl_2$ was added into the solution to make a lump. Fourth, 1 g of purified and oxidized bagasse was taken in 10 mL of multi-enzyme solution. Fifth, 0.5 g of sodium alginate and 0.5 g of bagasse was taken in 10 mL of multi-enzyme solution and dissolved. Then 10 mL of 0.2 M $CaCl_2$ was added into the solution to make a lump. Sixth, 1 g of purified and oxidized rice bran was taken in 10 mL of multi-enzyme solution. Seventh, 0.5 g of sodium alginate and 0.5 g of rice bran was taken in 10 mL of multi-enzyme solution and dissolved. Then 10 mL of 0.2 M $CaCl_2$ was added into the solution to make a lump. Seven different types of samples were incubated in a shaking incubator at 37°C for 1 h with constant shaking. After 1 hour, the lump was filtered and washed with distilled water for two times and was taken into seven different syringes.

## 2.11 Activity of different fixed bed bioreactor systems

For each fixed bed bioreactor, 10 mL corn starch solution (1%) was passed (flow rate 2 mL/min) through different bioreactors. Fixed bed bioreactors contained immobilized multi-enzyme in alginate or cellulosic materials or their combination. Then products from each fixed bed bioreactor were collected into different test tubes. Products (1.0 mL) from all bioreactors were taken in different test tubes. Resorcinol solution (2 mL) was added into all test tubes. The solutions containing test tubes were incubated at 80°C for 10 min. The test tubes were cooled at room temperature and then the absorbance (optical density) of all solution was measured at 520 nm. Instead of immobilized enzyme, free enzyme was control.

## 2.12 Reusability test of immobilized multi-enzyme in different fixed bed bioreactors

After completion of the first cycle, 10 mL distilled water flowed through the different bioreactors for washing. Then 10 mL of starch (1%) was flowed (flow rate 2 mL/min) through the different bioreactors. The product was collected from all bioreactors into different new test tubes. The fructose present in various test tubes was measured at 520 nm wavelength after resorcinol treatment. After completion of the second cycle, the above process was repeated for several cycles.

## 2.13 Statistical analysis

The mean values and standard deviations were calculated from three independent experiments. Statistical analysis was performed using one-way analysis of variance (ANOVA). The values obtained in the experiments were considered to be statistically significant, if $P < 0.05$.

## 3. Results and discussion

### 3.1 Different types of support matrices

Different types of multi-enzyme immobilized support matrices were prepared using various enzymes (α-amylase, glucoamylase and glucose isomerase) and support materials.

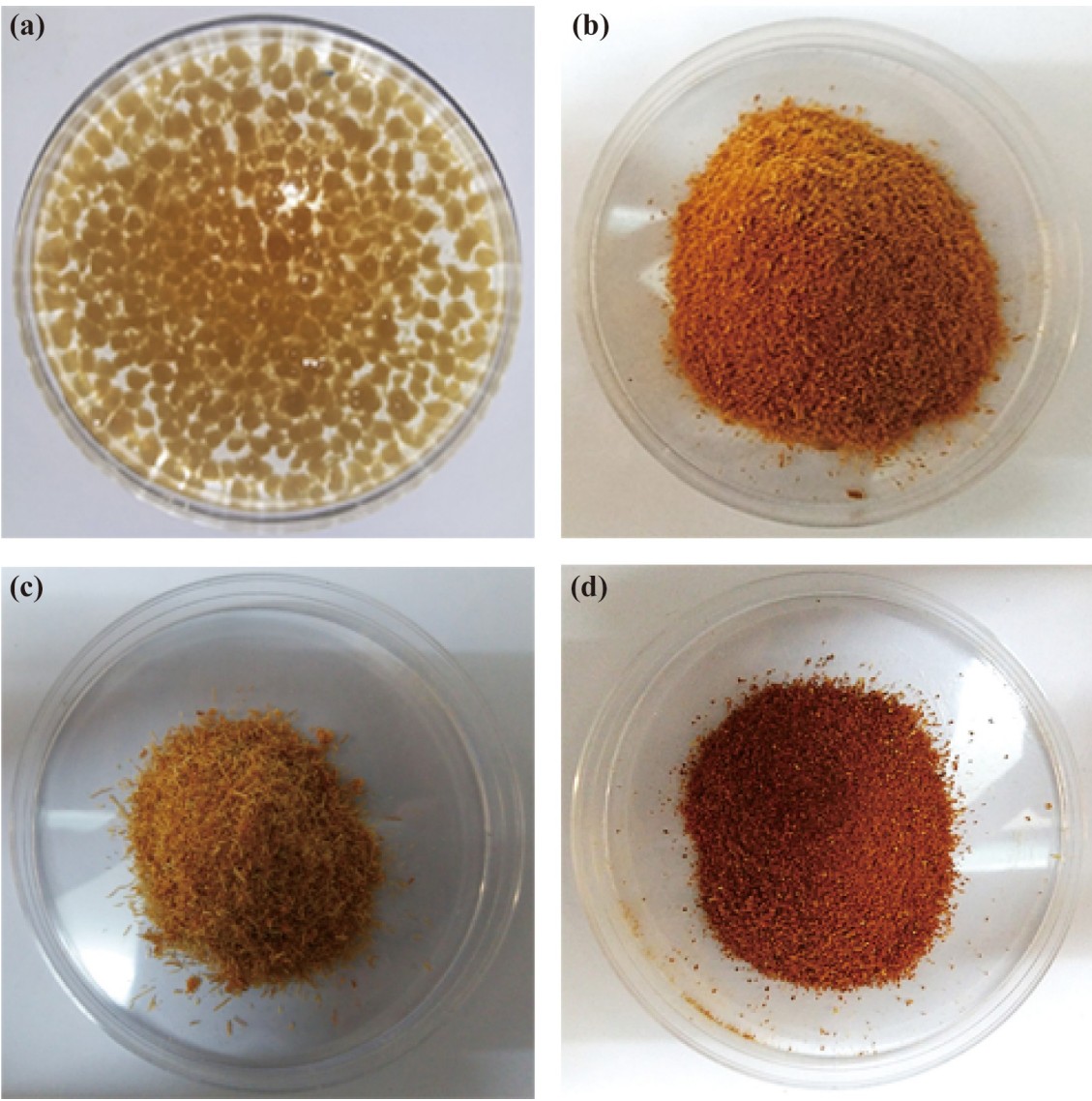

**Fig 1. Different types of support matrices which were used for the immobilization of multi-enzyme.** (a) alginate beads (b) sawdust (c) sugarcane bagasse and (d) rice bran.

Different types of multi-enzyme immobilized support matrix which were used for conversion of corn starch to HFCS. We were also prepared alginate (5%) beads with enzyme or without enzyme. Fig 1 shows the image of different types of support matrix which were used for the immobilization of multi-enzymes (α-amylase, glucoamylase and glucose isomerase).

The cellulosic materials (sawdust, sugarcane bagasse and rice bran) were treated with periodic acid. The materials were oxidized for the immobilization of multi-enzyme. The oxidation process helps to increase the binding of enzymes with support materials. Scanning Electron Microscopic (SEM) images of different cellulosic materials were taken before and after the oxidation process (Fig 2).

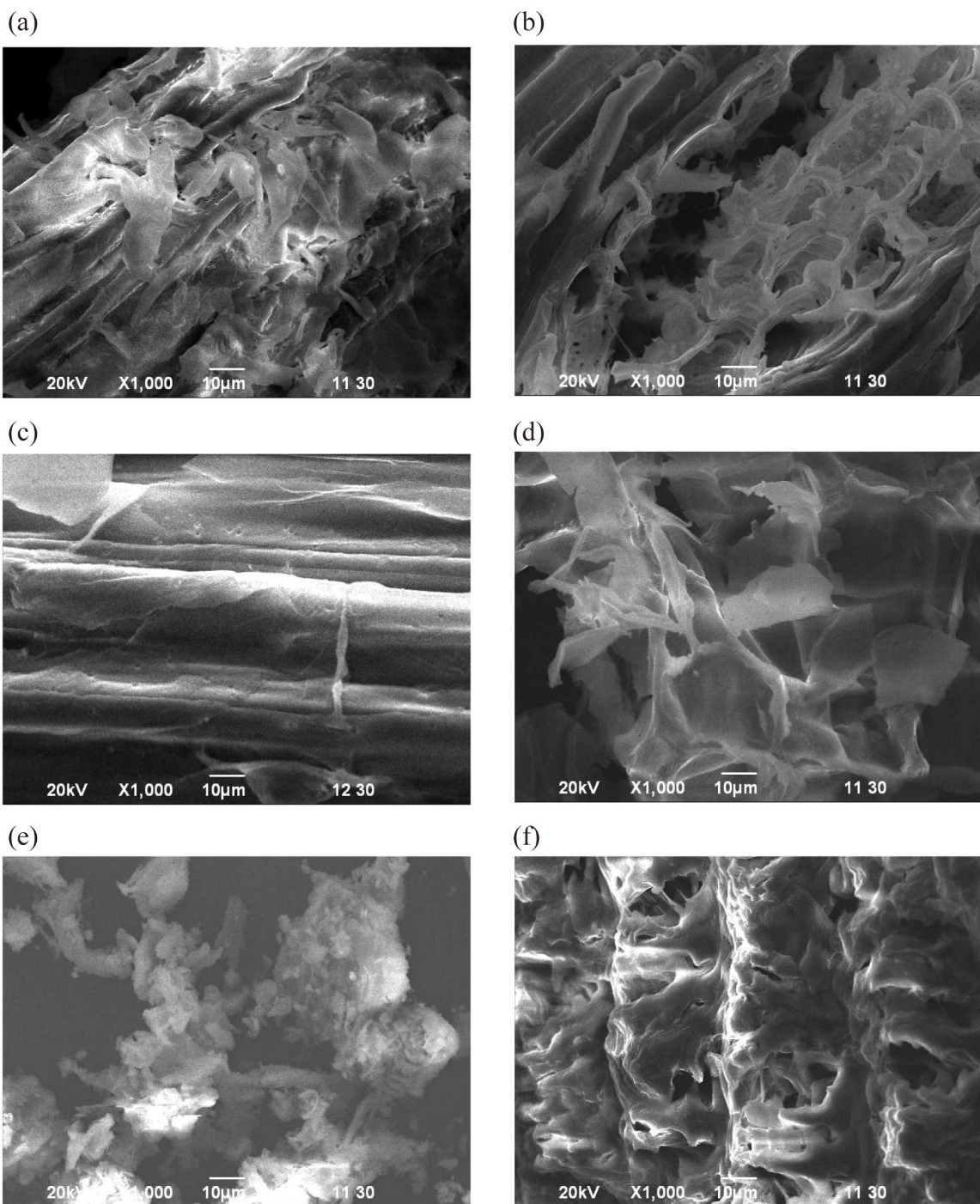

**Fig 2.** Scanning Electron Microscopic (SEM) images of different cellulosic materials before and after the oxidation process (a) sawdust before treatment, (b) sawdust after treatment, (c) sugarcane bagasse before treatment, (d) sugarcane bagasse after treatment, (e) rice bran before treatment and (f) rice bran after treatment.

## 3.2 Activity of single enzyme immobilization, multi-enzyme immobilization in sodium alginate beads, and free enzymes

The enzymatic activities of immobilized enzymes were determined through multi-enzyme immobilization process, single enzyme immobilization process and free enzyme. Among the three processes such as immobilized single enzyme, multi-enzyme, and free enzyme process, the sample absorbance at 520 nm were 0.566, 0.553 and 0.922 respectively. The free enzyme produced the highest fructose (Fig 3).

Free enzymes produced more fructose than immobilized enzymes (Fig 3). However, after one reaction cycle, free enzymes were not possible to use again. In case of multi-enzyme immobilization process and single-enzyme immobilization process, single enzyme immobilization showed slightly better results than multi-enzyme. But in the single enzyme immobilization process it took more time than the multi-enzyme immobilization process and also required three times more alginate beads. Hence, among them the multi-enzyme immobilization process was more suitable and efficient than other processes.

## 3.3 Reusability test of enzyme immobilized in sodium alginate beads (on the basis of enzyme activity)

Reusability is one of the most important parameters of enzyme immobilization. Enzymes immobilized in sodium alginate beads were reused up to eighth cycles. The result of free enzymes was good; however, free enzymes could not be used for more than one cycle. Thus, wastage of enzymes occurred due to incapability of reuse. On the contrary, immobilized multi-enzymes were used in several cycles. The result shows that the conversion rate of immobilized enzymes of the first cycle was lower than free enzymes. As the immobilized enzymes were used for several cycles; the sum of the results of total cycles were higher than the free

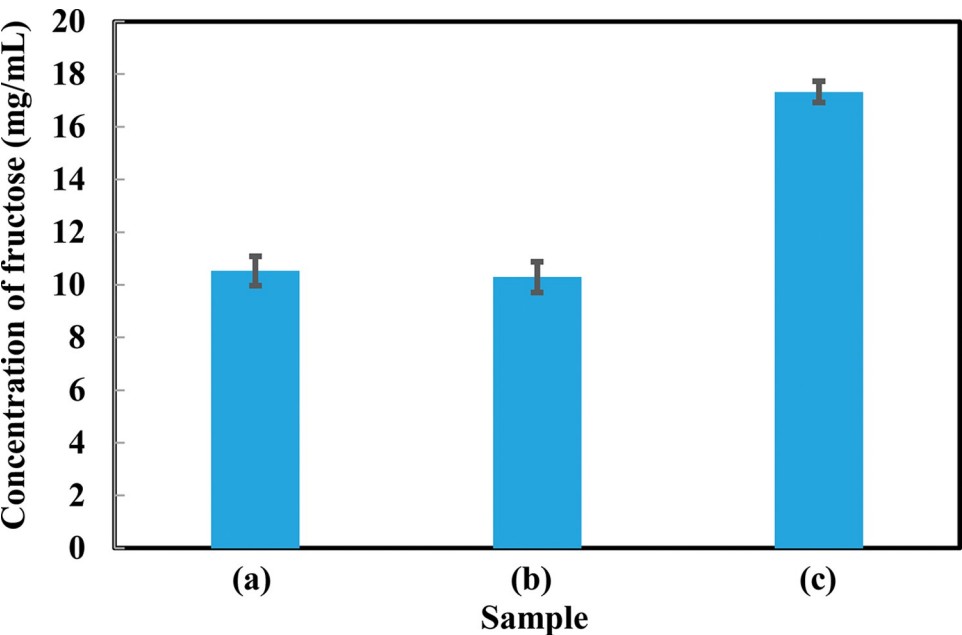

**Fig 3.** Fructose production in a single cycle of (a) single enzyme immobilization process (b) multi-enzyme immobilization process and (c) free enzymes. ***$p < 0.001$. The mean values and standard deviations (N = 3) are shown.

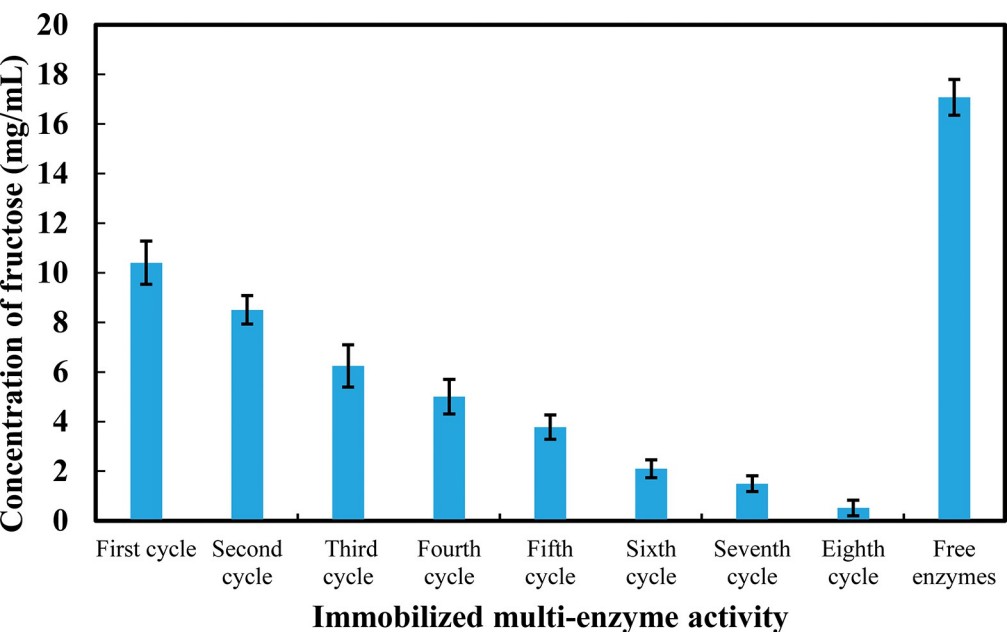

**Fig 4. Reuse of immobilized multi-enzyme sodium alginate beads in different cycles.** ***p < 0.001. The mean values and standard deviations (N = 3) are shown.

enzymes. Besides this, the product separation from immobilized enzymes was also easier than the free enzymes and wastage of enzymes were greatly reduced. The results are shown in Fig 4.

It was found that the enzymatic activities of multi-enzyme immobilized sodium alginate beads were very good up to fourth cycles. In the fifth cycle, the activity was moderate. In the sixth and seventh cycle, the enzymatic activities were very low. In every cycle, the activity of immobilized enzymes decreased gradually. This might have occurred due to breakage of the beads and leakage of enzymes in the washing steps in every cycle. Total yield of immobilized multi-enzymes was (10.40 + 8.51 + 6.25 + 5.00 + 3.77 + 2.10 + 1.49 + 0.52) mg/mL = 38.04 mg/mL of fructose. Free enzymes produced 17.07 mg/mL of fructose for only one time. Therefore, the overall outcome of immobilized enzymes in sodium alginate were better than the free enzymes.

### 3.4 Multi-enzyme immobilization in different cellulosic materials

Different cellulosic materials such as sawdust, sugarcane bagasse and rice bran were used to immobilize multi-enzyme (α-amylase, glucoamylase and glucose isomerase). It was observed that good quantity enzymes were immobilized in sawdust, bagasse and rice bran after acid treatment and the oxidation process (Fig 5). Enzymes were immobilized less amount into sawdust, bagasse and rice bran before acid treatment and oxidation process because the pore size of cellulosic material was small and other chemical substances were present that blocked the enzymes and decreased the enzymatic activity. The results are shown in Fig 5.

### 3.5 Reusability of immobilized multi-enzyme in cellulosic materials

Multi-enzyme (α-amylase, glucoamylase and glucose isomerase) were immobilized in different cellulosic materials such as sawdust, sugarcane bagasse and rice bran. It was found that good quantity enzymes were immobilized in sawdust and sugarcane bagasse (Fig 6). Enzymes were immobilized into small amounts in rice bran. Enzymes immobilized in every cellulosic

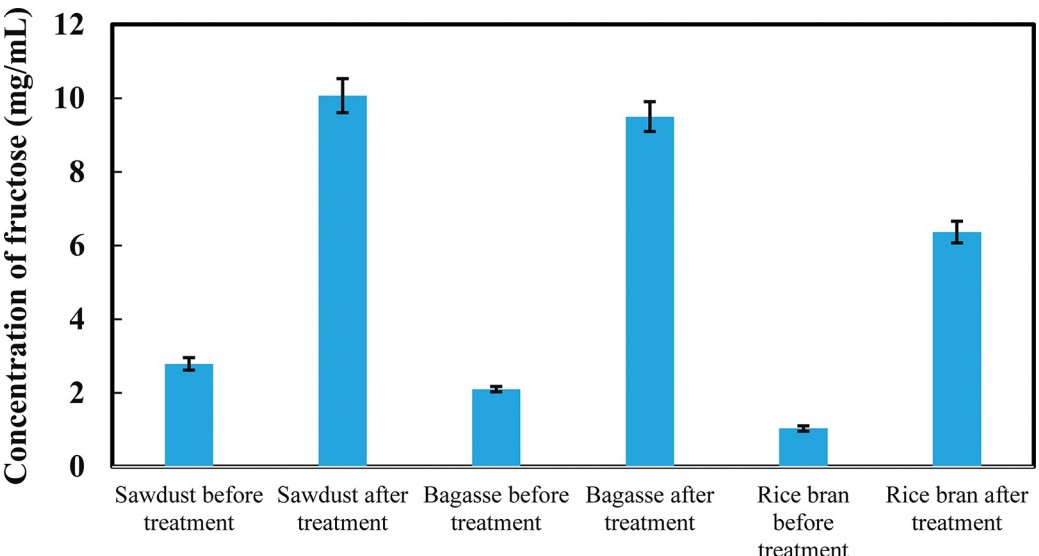

**Fig 5. The effects of acid and oxidation treatment on different cellulosic materials (sawdust, sugarcane bagasse and rice bran).** ***p < 0.001. The mean values and standard deviations (N = 3) are shown.

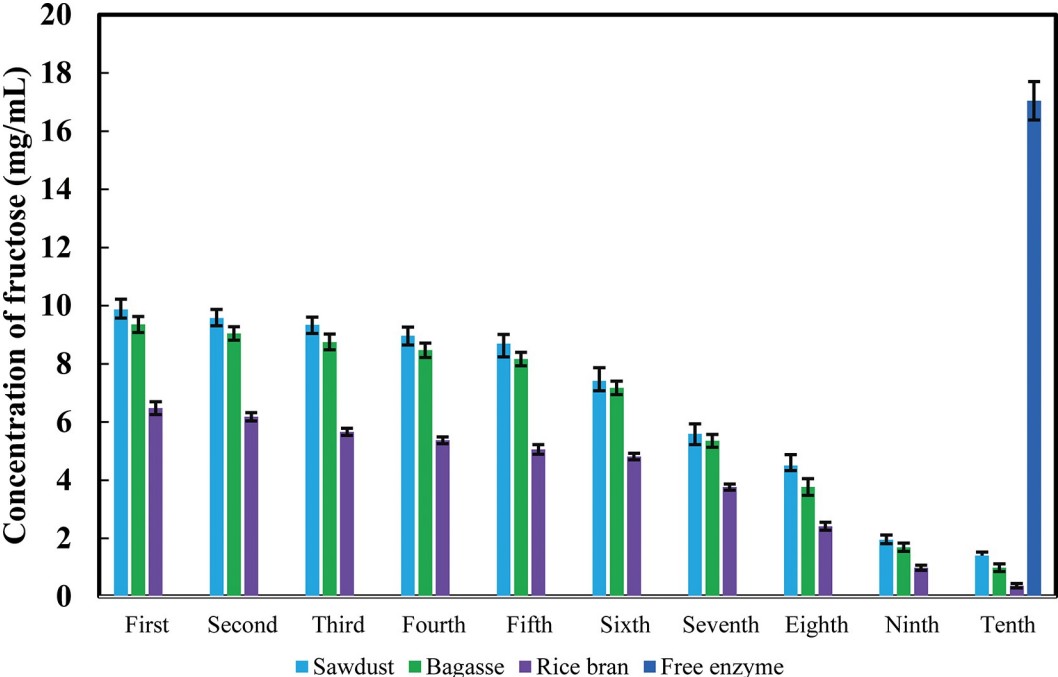

**Fig 6. Various cycle activities of immobilized multi-enzyme in cellulosic materials.** ***p < 0.001. The mean values and standard deviations (N = 3) are shown.

material were reused for up to ten cycles. Enzymes were immobilized in sawdust, and bagasse showed good enzymatic activities up to six cycles. After six cycle enzyme activity was greatly decreased. In every cycle enzyme activity decreased due to enzyme leakage. The free enzymes used only in the first cycle.

### 3.6 Activity of immobilized enzymes in different bioreactors

Various types of support matrix were used for immobilizing the enzymes and to develop a fixed bed bioreactor system for the production of HFCS from corn starch. The productivity of immobilized multi-enzymes in different types of fixed bed bioreactor were shown in Fig 7. The fixed bed enzyme bioreactors which contained immobilized enzymes in sawdust + alginate conjugate showed the heights activity that means this bioreactor converts the highest amount of fructose (10.88 mg/mL) from starch in the first cycle. The bioreactors with bagasse + alginate conjugate (10.16 mg/mL) and only sawdust (10.15 mg/mL) also showed very good results in the first cycle. In addition, enzyme bioreactors with only bagasse and only sodium alginate showed moderate results. The bioreactors with rice bran + alginate conjugate and only rice bran showed less activities. Free enzymes produced 17.28 mg/mL of fructose. The activities of free enzymes were always better than immobilized enzymes however free enzymes cannot be reused while immobilized enzymes were reused up to several cycles.

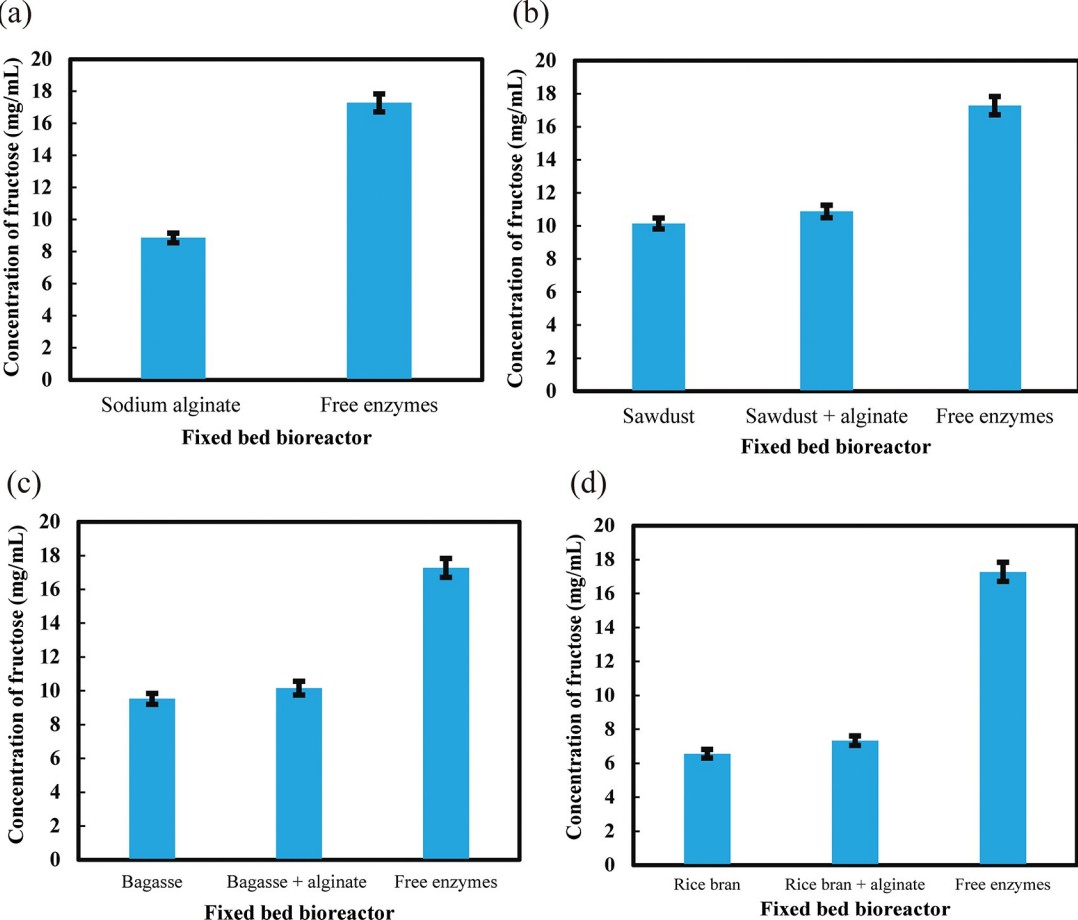

**Fig 7.** Production of fructose (mg/mL) from (a) sodium alginate fixed bed multi-enzyme bioreactor (b) sawdust fixed bed multi-enzyme bioreactor (c) bagasse fixed bed multi-enzyme bioreactor and (d) rice bran fixed bed multi-enzyme bioreactor. ***p < 0.001. The mean values and standard deviations (N = 3) are shown.

### 3.7 Reusability test of immobilized multi-enzyme in different bioreactors

All the bioreactors were used till the tenth cycle. The best result was found from sawdust + alginate (with enzymes) conjugate bioreactor. They produced 84.82 mg/mL of total fructose. All multi-enzyme immobilized bioreactors produced a good amount of total fructose. Free enzymes produced 17.25 mg/mL of fructose, but it was not possible to use again and therefore it was wasted. But all the bioreactors were reused and produce more total fructose than free enzymes (Table 1).

Fixed bed bioreactors were used for several cycles for the conversion of starch to fructose. They showed satisfactory results. Though in every cycle enzyme activity decreased due to leakage of some enzymes. In all bioreactor fructose production was very good from the first to the seventh cycle. After the seventh cycle, the rate of fructose production started to decrease. In the tenth cycle the rate decreased drastically (Table 1). After the tenth cycle, all the bioreactors showed zero results due to leakage of all immobilized enzymes.

Different concentrations of sodium alginate were used and found that the 5% alginate concentration was best. Single enzyme immobilization process showed quite better results than the multi-enzyme. But a multi-enzyme immobilization process requires less time and a smaller amount of sodium alginate than a single enzyme immobilization process. Therefore, among them, the multi-enzyme immobilization process was more reliable than other processes. The reusability test indicates that free enzymes can be used only one cycle and after one cycle it cannot be used. However, multi-enzymes immobilized matrices were used in several cycles. The total amounts of fructose production (mg/mL) from starch solution using immobilized multi-enzyme matrices were higher than the free enzymes.

Here, some cheap cellulosic materials were used for the immobilization of α-amylase, glucoamylase and glucose isomerase enzymes. The cellulosic materials need to perform an oxidation process. This oxidation process added an extra aldehyde group to the cellulosic materials that covalently bind with the amino group of enzymes [20]. Here, sawdust, sugarcane bagasse and rice bran were oxidized with 0.03 M periodic acid and it acts as a good binder in the multi-enzyme immobilization process. Immobilized enzymes in these materials showed better results than only sodium alginate and free enzymes. Multi-enzyme immobilization in these cellulosic

**Table 1. Reusability of immobilized multi-enzyme in different fixed bed enzyme bioreactors.**

| Cycle | Concentration of fructose (mg/mL) | | | | | | | |
|---|---|---|---|---|---|---|---|---|
| | (a) | (b) | (c) | (d) | (e) | (f) | (g) | (h) |
| 1st | 8.89 | 10.14 | 10.80 | 9.52 | 10.16 | 6.56 | 7.39 | 17.25 |
| 2nd | 8.38 | 9.83 | 10.33 | 9.10 | 9.73 | 6.12 | 6.84 | |
| 3rd | 8.04 | 9.48 | 9.73 | 8.86 | 9.38 | 5.76 | 6.47 | |
| 4th | 7.50 | 9.13 | 9.41 | 8.42 | 9.13 | 5.58 | 6.06 | |
| 5th | 7.25 | 8.80 | 8.96 | 7.98 | 8.59 | 5.38 | 5.67 | |
| 6th | 6.96 | 8.38 | 8.75 | 7.65 | 8.02 | 5.07 | 5.36 | |
| 7th | 6.09 | 7.69 | 8.25 | 6.67 | 6.95 | 4.24 | 4.83 | |
| 8th | 5.15 | 7.08 | 7.48 | 5.67 | 5.81 | 3.73 | 3.83 | |
| 9th | 4.38 | 5.83 | 6.45 | 4.70 | 5.11 | 2.28 | 2.49 | |
| 10th | 2.69 | 4.38 | 4.66 | 2.85 | 2.99 | 1.18 | 1.53 | |
| **Total** | **65.33** | **80.74** | **84.82** | **71.42** | **75.87** | **45.90** | **50.47** | **17.25** |

**Note:** (a) Sodium alginate (with enzymes), (b) Sawdust (with enzymes), (c) Sawdust + alginate (with enzymes), (d) Bagasse (with enzymes), (e) Bagasse + alginate (with enzymes), (f) Rice bran (with enzymes), (g) Rice bran + alginate (with enzymes) and (h) Free enzymes

***$p < 0.001$. The mean values (N = 3) are shown.

materials can be used for several cycles than sodium alginate because sodium alginate beads are fragile. Immobilization enzymes in different cellulosic materials can create a new window in the field of enzyme technology. On the other hand, when cellulosic materials were mixed with sodium alginate the results were found better than using a single material as a matrix.

## 4. Conclusion

Enzymes are biological molecules that act as a catalyst and speed up the reactions by decreasing the activation energy without any change in their structure. Generally, most of the enzymes are comparatively unusable after reaction due to the high separation costs after use. Immobilized enzymes have great potential in the field of biotechnology. Multi-enzyme immobilization is a process in which different types of enzymes are immobilized in a single matrix. In this study, multi-enzyme such as α-amylase, glucoamylase and glucose isomerase were immobilized in various types of support matrices like sodium alginate, sawdust, sugarcane bagasse, rice bran and combination of alginate with cellulosic materials. The activities of the immobilized multi-enzyme system for the production of HFCS from the starch solution were determined. Multi-enzyme (α-amylase, glucoamylase and glucose isomerase) immobilized in sodium alginate showed very good results. Immobilization of α-amylase, glucoamylase and glucose isomerase enzymes in sawdust, sugarcane bagasse and rice bran showed better results than sodium alginate and free enzymes. Multi-enzyme immobilized in these materials can be used for several cycles than sodium alginate because sodium alginate beads are fragile. On the other hand, when cellulosic materials were mixed with sodium alginate, the results were better than using a single material as a matrix. A fixed bed immobilized multi-enzyme bioreactor had been established which could be used for the production of HFCS in the industrial scale. The whole research study was carried on focusing on industrial applications. Immobilization of enzymes in these materials can create a new casement in the field of enzyme technology. Therefore, the present study could create a new opportunity for the application of immobilized multi-enzyme systems in many areas of industrial biotechnology.

## Supporting information

**S1 Data.**
(XLSX)

## Author Contributions

**Conceptualization:** Md. Zahidul Islam.

**Data curation:** Md. Zahidul Islam.

**Formal analysis:** Sabbir Janee, Shatabdy Saha, Sabrina Sharmin, A. Q. Fuad Hasan, Umme Salma Zohora, Ripa Moni, Md. Zahidul Islam.

**Investigation:** Sabbir Janee, Shatabdy Saha, Sabrina Sharmin, A. Q. Fuad Hasan, Umme Salma Zohora, Ripa Moni.

**Methodology:** Md. Zahidul Islam, Mohammad Shahedur Rahman.

**Writing – original draft:** Sabbir Janee, Sabrina Sharmin, Umme Salma Zohora, Md. Zahidul Islam, Mohammad Shahedur Rahman.

**Writing – review & editing:** Sabrina Sharmin, Umme Salma Zohora, Md. Zahidul Islam, Mohammad Shahedur Rahman.

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
