## [Decision Letter · Decision Letter 0]

25 Aug 2023

PONE-D-23-22990Construction and Investigation of Multi-enzyme Immobilized Matrix for the Production of HFCSPLOS ONE

Dear Dr. lslam,

Thank you for submitting your manuscript to PLOS ONE. After careful consideration, we feel that it has merit but does not fully meet PLOS ONE’s publication criteria as it currently stands. Therefore, we invite you to submit a revised version of the manuscript that addresses the points raised during the review process.

We look forward to receiving your revised manuscript.

Kind regards,

Saleh Ahmed Mohamed

Academic Editor

PLOS ONE

“The funders had no role in study design, data collection and analysis, decision to publish, or preparation of the manuscript”

3. "In your Data Availability statement, you have not specified where the minimal data set underlying the results described in your manuscript can be found. PLOS defines a study's minimal data set as the underlying data used to reach the conclusions drawn in the manuscript and any additional data required to replicate the reported study findings in their entirety. All PLOS journals require that the minimal data set be made fully available. For more information about our data policy, please see http://journals.plos.org/plosone/s/data-availability.

Additional Editor Comments:

The authors should be revised the manuscript based on reviewer comments.

Reviewers' comments:

Reviewer's Responses to Questions

**Comments to the Author**

1. Is the manuscript technically sound, and do the data support the conclusions?

Reviewer #1: Yes

2. Has the statistical analysis been performed appropriately and rigorously? 

Reviewer #1: No

3. Have the authors made all data underlying the findings in their manuscript fully available?

Reviewer #1: Yes

4. Is the manuscript presented in an intelligible fashion and written in standard English?

Reviewer #1: Yes

5. Review Comments to the Author

Reviewer #1: The manuscript is suitable for publication after minor revision:

- Figure 3: in column ligands written 1, 2 and 3 codes for processes but in fig. description the author write a, b and c. In single enzyme hydrolysis, please clarify whether the process done in 3 steps or all enzymes together in one step

- All result lack the statistical analysis such as ANOVA

- Please improve the resolution of all images

6. PLOS authors have the option to publish the peer review history of their article (what does this mean?). If published, this will include your full peer review and any attached files.

Reviewer #1: No

---

## [Author Response · Author response to Decision Letter 0]

9 Sep 2023

Re: PLOS ONE Decision: Revision required [PONE-D-23-22990] - [EMID:9cad34958b1a9769]

Title: " Construction and Investigation of Multi-enzyme Immobilized Matrix for the Production of HFCS "

 Author(s): Sabbir Janee1, Shatabdy Saha1, Sabrina Sharmin2, A.Q. Fuad Hasan1, Umme Salma Zohora1, Ripa Moni1, Md. Zahidul Islam1,# and Mohammad Shahedur Rahman1,3

September 10, 2023

To,

Saleh Ahmed Mohamed

Academic Editor

PLOS ONE

Dear Sir,

Thank you very much for your letter on our manuscript (PONE-D-23-22990) dated on 21 July 2023. In accordance with the reviewers’ comments, we made the revised version of the manuscript. In the manuscript with highlights, the revised portions according to the reviewers are indicated by red letters. We have provided the minimal underlying data as an Excel file from which we have derived our interpretation of the study. Would you please take care of the revised manuscript? 

Thank you very much.

Sincerely yours,

Dr. Md. Zahidul Islam

Associate Professor 

Department of Biotechnology and Genetic Engineering, 

Jahangirnagar University, Savar, Dhaka-1342, Bangladesh.

Phone: +88-01787-086965

E-mail: zahidul@juniv.edu

Answer to the Journal requirements

First of all, we wish to appreciate for your valuable comments and suggestions. We revised our manuscript according to the comments and answers to the given questions are as follows.

Thank you very much for your comment. According to the comment, we have revised our manuscript in pursuance of the PLOS ONE style templates.

“The funders had no role in study design, data collection and analysis, decision to publish, or preparation of the manuscript”

In this research project, we did not receive any funds or grants.

We did not receive any funds from any funder. Therefore, the funders had no role in study design, data collection and analysis, decision to publish, or preparation of the manuscript.

Not Applicable

The authors received no specific funding for this work.

3. "In your Data Availability statement, you have not specified where the minimal data set underlying the results described in your manuscript can be found. PLOS defines a study's minimal data set as the underlying data used to reach the conclusions drawn in the manuscript and any additional data required to replicate the reported study findings in their entirety. All PLOS journals require that the minimal data set be made fully available. For more information about our data policy, please see http://journals.plos.org/plosone/s/data-availability.

We have provided the minimal underlying data as an Excel file from which we have derived our interpretation of the study.

According to the comment, we have updated the references using EndNote software.

Answer to the Reviewer-1 comments:

First of all, we wish to appreciate for your valuable comments and suggestions. Here is our reply to your comments.

1. Is the manuscript technically sound, and do the data support the conclusions?

Reviewer #1: Yes

Thank you very much for your comment.

2. Has the statistical analysis been performed appropriately and rigorously?

Reviewer #1: No

According to the comment, we have analyzed all data statistically using one-way analysis of variance (ANOVA). We have revised the content as follows. 

(Page-14)

Statistical analysis: 

The mean values and standard deviations were calculated from three independent experiments. Statistical analysis was performed using one-way analysis of variance (ANOVA). The values obtained in the experiments were considered to be statistically significant, if P < 0.05.

3. Have the authors made all data underlying the findings in their manuscript fully available?

Reviewer #1: Yes

Thank you very much for your comment.

4. Is the manuscript presented in an intelligible fashion and written in standard English?

Reviewer #1: Yes

Thank you very much for your valuable comment on our manuscript.

5. Review Comments to the Author

Reviewer #1: The manuscript is suitable for publication after minor revision:

- Figure 3: in column ligands written 1, 2 and 3 codes for processes but in fig. description the author write a, b and c. 

According to the comment, we have revised the figure 3. 

In single enzyme hydrolysis, please clarify whether the process done in 3 steps or all enzymes together in one step

According to the comment, we have revised the content as follows. 

(Page-9)

Determination of activity of different enzyme immobilization systems

For single-enzyme immobilization, each matrix material (Immobilized single-enzymes in sodium alginate) were taken in three different falcon tubes. 10 mL of corn starch solution (1%) was added in the tube containing the α-amylase immobilized matrix and incubated for 30 min at 37 ºC. Then the solution was collected and added in the second tube containing the glucoamylase immobilized matrix and incubated for 30 min at 37 ºC. Then the solution was collected from the second tube and added in the third tube containing the glucose isomerase immobilized matrix and incubated for 30 min at 37 ºC. In case of multi-enzymes immobilization, the immobilized multi-enzyme in sodium alginate beads were taken in a falcon tube and 10 mL of corn starch solution (1%) was added in the tube. For the control experiment, 10 mL of corn starch solution (1%) and 10 mL free enzyme were taken in a falcon tube. All the tubes were incubated for 30 min at 37 ºC. Then 1 mL solution from each tube was taken into new test tubes, and 2 mL of resorcinol solution was added and incubated all test tubes at 80 ºC for 10 min in a water bath. The solution temperature was cooled at room temperature and optical density (OD) of control and sample solutions were measured at 520 nm wavelength using UV-Vis spectrophotometer. The productivity of fructose (mg/mL) was determined from a standard calibration curve. To determine the productivity of fructose by an immobilized enzyme in support matrices and free enzyme, three independent experiments were carried out for each matrix. Average values and standard deviations of productivity of immobilized enzymes in support matrices and free enzymes were calculated.

- All result lack the statistical analysis such as ANOVA

According to the comment, we have analyzed all results statistically using one-way analysis of variance (ANOVA). We also have revised all figures after reanalyzed all results statistically.

- Please improve the resolution of all images

According to the comment, we have revised all figures with higher resolution images.

6. PLOS authors have the option to publish the peer review history of their article (what does this mean?). If published, this will include your full peer review and any attached files.

Do you want your identity to be public for this peer review? For information about this choice, including consent withdrawal, please see our Privacy Policy.

Reviewer #1: No

Thank you very much for your comment.

---

## [Editor Report · Decision Letter 1]

2 Oct 2023

Construction and Investigation of Multi-enzyme Immobilized Matrix for the Production of HFCS

PONE-D-23-22990R1

Dear Dr.  Zahidul lslam

We’re pleased to inform you that your manuscript has been judged scientifically suitable for publication and will be formally accepted for publication once it meets all outstanding technical requirements.

Kind regards,

Saleh Ahmed Mohamed

Academic Editor

PLOS ONE

Additional Editor Comments (optional):

The revised manuscript has been accepted

---

## [Editor Report · Acceptance letter]

26 Oct 2023

PONE-D-23-22990R1 

Construction and Investigation of Multi-enzyme Immobilized Matrix for the Production of HFCS 

Dear Dr. Islam:

I'm pleased to inform you that your manuscript has been deemed suitable for publication in PLOS ONE. Congratulations! Your manuscript is now with our production department. 

Kind regards, 

on behalf of

Dr. Saleh Ahmed Mohamed 

Academic Editor

PLOS ONE